# Evaluation of Preosteoblast MC3T3-E1 Cells Cultured on a Microporous Titanium Membrane Fabricated Using a Precise Mechanical Punching Process

**DOI:** 10.3390/ma13225288

**Published:** 2020-11-22

**Authors:** Jingyu Zhang, Yukihiko Sakisaka, Hiroshi Ishihata, Kentaro Maruyama, Eiji Nemoto, Shigeki Chiba, Masaru Nagamine, Hiroshi Hasegawa, Satoru Yamada

**Affiliations:** 1Division of Periodontology and Endodontology, Department of Oral Biology, Tohoku University Graduate School of Dentistry, 4-1 Seiryo-machi, Aoba-ku, Sendai 980-8575, Japan; zhangjingyuzhudi@163.com (J.Z.); yuki@dent.tohoku.ac.jp (Y.S.); maruyama.k@dent.tohoku.ac.jp (K.M.); e-nemoto@dent.tohoku.ac.jp (E.N.); satoruy@tohoku.ac.jp (S.Y.); 2Nagamine Manufacturing Co., Ltd., 1725-26, Kishinoue, Manno-town, Nakatado-Gun, Kagawa-prefecture 766-0026, Japan; s_chiba@nagamine-manu.co.jp (S.C.); nagakatu-nrd@me.pikara.ne.jp (M.N.); 3Department of Oral Surgery and Dentistry, Fukushima Medical University, 1, Hikariga-oka, Fukushima 960-1295, Japan; hasehi@fmu.ac.jp

**Keywords:** osteogenic differentiation, calcification, MC3T3-E1 cell, microporous titanium, titanium topography, dentistry

## Abstract

The surface topography of Titanium (Ti) combined toughness and biocompatibility affects the attachment and migration of cells. Limited information of morphological characteristics, formed by precise machining in micron order, is currently available on the Ti that could promote osteoconduction. In the present study, a pure Ti membrane was pierced with precise 25 μm square holes at 75 μm intervals and appear burrs at the edge of aperture. We defined the surface without burrs as the “Head side” and that with burrs as the “Tail side”. The effects of the machining microtopography on the proliferation and differentiation of the preosteoblasts (MC3T3-E1 cells) were investigated. The cells were more likely to migrate to, and accumulate in, the aperture of holes on the head side, but grew uniformly regardless of holes on the tail side. The topography on the both surfaces increased osteopontin gene expression levels. Osteocalcin expression levels were higher on the head side than one on the blank scaffold and tail side (*p* < 0.05). The osteocalcin protein expression levels were higher on the tail side than on the head side after 21 days of cultivation, and were comparable to the proportion of the calcified area (*p* < 0.05). These results demonstrate the capacity of a novel microporous Ti membrane fabricated using a precise mechanical punching process to promote cell proliferation and activity.

## 1. Introduction

Titanium (Ti) is a commonly used biomaterial in the medical field because of its superior biocompatibility and promotion of the osteoconductive properties of living bone tissue [1], as well as its high weight tolerance of the physical loads placed on dental and orthopedic bone [2,3]. The bioinert surface of the Ti oxide layer allows for the assembly of tissue repair components for the osteogenic restoration of osseointegration. The successful tissue remodeling of natural bone devices, with implants, results in uninterrupted contact between the surface of Ti and adjacent bone [4]. Ti is generally superior to ceramics and polymers for application to implantable devices, because, pure Ti or Ti alloy scaffolds, which are transplanted onto recipient sites of human living tissue, have high weight tolerance, fracture toughness, and fatigue deformation [5]. The morphological properties of the surface of Ti that contribute to its mechanical performance and biocompatibility have been enhanced by the development of microfabrication technology [6].

With the rapid advances in research on biomaterials, a number of technological approaches have been applied to biomaterials in order to modify their morphology, porosity, and topography. Titanium is currently used in dentistry for implants [7] and miniscrews [8], showing excellent mechanical and biological properties. Novel dental implant products employ surface modifications on a microtopographical scale, even a nanotopographical scale, which promote the attachment and migration of cells from the surrounding living tissue at the recipient site [9,10]. The modified surface topography of biomaterials promotes osseointegration and osteoconduction [11,12]. Microfabrication, using laser ablation, is a commonly used approach to modify the microporous surface of Ti. However, laser beam irradiation melts the surface of Ti, resulting in chemical composition changes due to the Ti oxide matrix produced by the solidification of the melted Ti [13].

In the present study, commercially pure Ti was fabricated to prepare micropore arrays using a microarrayed pin needle punch (MPNP) in preventing a thermal effect on the process. We investigated a behavior of bone tissue-derived cells [14] cultivated on the Ti surface including the micropore arrays and analyzed characteristics of a proliferation and differentiation of the cells.

## 2. Materials and methods

### 2.1. Cross-Cut Preparation of MPNP

The MPNP tool was prepared as follows: On the plane surface of a metal block of a cemented carbide blank (Super ultra-fine grained carbide “AF1”: Sumitomo Electric Hardmetal, Itami, Japan), grooves with a width of 50 μm were serially carved by a grinding process at a precise equidistant of 75 μm using a diamond dicing wheel. The processed surface of the block was then horizontally rotated by 90 degrees and subjected to carving, using the same process to make grooves, with a direction that was perpendicular to the array of grooves already formed. This process was performed using a fully automatic dicing saw (UPG310Li: Okamoto Machine Tool Works Ltd., Annaka, Japan) and produced lattice arrayed needle punches that were precisely formed as uniform prism-shaped pins and the square side at the tip of the pins had a length of 25 μm. The square surface of the punch tool with a microneedle array included approx. 17,700 pins within a 10 mm square (Figure 1).

### 2.2. Fabrication of a Microperforated Ti Membrane

A thin pure Ti plane sheet with a thickness of 10 μm was used to fabricate material with a precise microporous structure. The fabrication process with MPNP was performed for piercing through the Ti sheet by cold working (Figure 2). Prior to punching, a hard resin flat plate, capable of plastic deformation, was positioned on the target section of the work table, and MPNP tool was pressed into the plate and imprinted to allow the resin to function as a die in punching. Following the preservation of the positions of the punch and the engraving surface of the die, a Ti sheet was placed between the punch and the die. A servo press puncturing (1500 kg/cm^2^) to the workpiece of the Ti sheet generated pores in a uniform manner. All hole shapes in a single perforation process were uniform. After the release of the punch from the workpiece, precise square holes transferred from the needle shape had formed in all areas of the impacted surface under the punch. However, volcanic-shaped apertures appeared at the exit side of the pierced sheet. We defined the flat side as the “head side” and the volcanic side as the “tail side”. Each microporous area was cut in a circle with a diameter of 22 mm.

### 2.3. Preparation of Processed Ti Specimens for Cultivation

Ti specimens were ultrasonically cleaned in 70% ethanol (Kenei Pharmaceutical Co., Ltd., Osaka, Japan) for 15 min, thoroughly rinsed in distilled water with ultrasonication for 15 min, and then dried and autoclaved. To increase surface hydrophilicity, all Ti specimens were treated with plasma processing using the Desktop Vacuum Plasma Processing System (Strex Inc., Osaka, Japan).

### 2.4. Cultivation of the Osteogenic Cell Line

The murine pre-osteoblastic cell line, MC3T3-E1 [14,15] was obtained from the American Type Culture Collection (Manassas, VA, USA) and maintained in α-Minimum Essential Medium (α-MEM) (Gibco^®^/Life Technologies, Carlsbad, CA, USA) containing 10% fetal bovine serum (FBS) (Biowest, Nuaillé, France). The cells were seeded at 50,000 cells/cm^2^ on 12-well culture plates with a microporous Ti membrane or non-microporous membrane (blank) and incubated. To compare the topography of two sides of the microporous Ti membrane, cells were seeded on Head or Tail side of it previously set on the well. To assess the proliferation and migration of cells on the membrane, cells were cultured in α-MEM with 2% FBS. To induce osteogenic differentiation, cells were cultured in 5% FBS α-MEM with or without 5 mM β-glycerophosphate (Sigma Chemical Co., St. Louis, MO, USA) and 50 μg/mL ascorbic acid (Sigma) (osteogenic medium). The medium was changed every three days.

### 2.5. Scanning Electron Microscope (SEM) Observations of Cell Morphology

Specimens cultured for 3 and 30 days in α-MEM with 2% FBS were fixed in 2% (*w/v*) glutaraldehyde in phosphate-buffered saline (PBS, Fujifilm Wako Pure Chemical Co., Osaka, Japan) and incubated at 4 °C for 1 h. After washing with PBS, specimens were dehydrated by washing in increasing concentrations of ethanol. Following immersion in t-butyl alcohol for 30 min, specimens were lyophilized with a critical point dryer (VFD21-S, VACUM DEVICE, Ibaraki, Japan). Dried samples were mounted on aluminum stages using double-sided tape and then coated with platinum using an ion sputter coater (JFC-1600, JEOL, Tokyo, Japan). All specimens were observed using SEM (JSM-6390LA, JEOL, Tokyo, Japan).

### 2.6. Immunofluorescence Assay

Specimens cultured in osteogenic medium for 7, 14 and 21 days were fixed with 2% (*w/v*) paraformaldehyde (Nacalai Tesque Inc., Kyoto, Japan) for 15 min. They were incubated with PBS, containing 0.25% Triton X-100 (Sigma) (PBS-T) for 10 min for permeabilization, and then incubated in PBS-T with 1% (*w/v*) bovine serum albumin (Sigma) for 30 min to block non-specific binding [16]. Specimens were incubated with a rat anti-osteocalcin (OCN) antibody (1:200 dilution, Takara Bio Inc., Kusatsu, Japan) and rabbit anti-osteopontin (OPN) antibody (1:1000 dilution, Abcam, Cambridge, MA, USA) for 1 h, followed by an incubation with an Alexa Fluor^®^ 488-conjugated goat anti-mouse secondary antibody (1:1000 dilution, Abcam) and Alexa Fluor^®^ 555-conjugated goat anti-rabbit secondary antibody (1:1000 dilution, Abcam) for 1 h. Specimens were also incubated with 4′,6-diamidino-2-phenylinole (DAPI) (Invitrogen^TM^/Life Technologies) diluted in PBS for 5 min to identify nuclei. Immunofluorescence images were evaluated using immunofluorescence microscopy (Leica M165FC, Leica Microsystems, Nussloch, Germany).

### 2.7. Reverse Transcription and Real-Time Quantitative Polymerase Chain Reaction (RT-PCR)

To harvest cells on the microporous area of the Ti membrane, the outside area was removed before homogenization. Total cellular RNA was extracted using Qiashredder with an RNeasy Kit (QIAGEN, Hilden, Germany) according to the manufacturer’s instructions, and was then treated with DNase (DNA-freeTM, Ambion^®^/Life Technologies). Then, total RNA was reverse-transcribed into first-strand complementary DNA (cDNA) using a Transcriptor First Strand cDNA Synthesis Kit^®^ (Roche Diagnostics GmbH, Mannheim, Germany) in accordance with the manufacturer’s protocol. cDNA, theoretically converted from 50 ng of total RNA, was used. All primers were designed using LightCycler probe design software^®^ (Roche Diagnostics GmbH), and the primer sequences for each target mouse gene encoding OCN, OPN and Glyceraldehyde 3-phosphate dehydrogenase (GAPDH) were as followed (forward/reverse): OCN (5′-TGAACAGACTCCGGCG-3′/5′-GATACCGTAGATGCGTTTG-3′); OPN (5′-TTTACAGCCTGCACCC-3′/5′-CTAGCAGTGACGGTCT-3′); and GAPDH (5′-AATGTGTCCGTCGTGGATCTGA-3′/5′-GATGCCTGCTTCACCACCTTCT-3′). The amplification profile in real-time PCR was 40 cycles at 95/3 and 60/20 [temperature (°C)/time (s)]. PCR was performed using the CFX96 TouchTM Real-Time PCR Detection System (Bio-Rad Laboratories, Hercules, CA, USA) with KAPA SYBR^®^ FAST (Kapa Biosystems, Boston, MA, USA) and optimized levels of 3 mM MgCl_2_ and 500 nM of each primer. After amplification, one cycle of a linear temperature gradient from 55 °C to 95 °C at a transition rate of 0.5 °C/30 s was performed to assess the specificity of the PCR products. In each run, water was used as the negative control. The relative expression levels of transcripts are shown after normalization to the corresponding sample expression level of GAPDH.

### 2.8. Detection and Quantification of Mineralization

Mineralization was detected and quantified using the Alizarin red S (ARS) staining assay. Specimens cultured in osteogenic medium for 28 days were fixed with 2% (*w/v*) paraformaldehyde in PBS for 15 min and rinsed thoroughly with distilled water. The specimens were then stained with 40 mM ARS (pH 4.2) at room temperature for 5 min. Specimens were rinsed repeatedly with distilled water to remove excess dye. Images of stained cells were obtained with Epson ES-2200 (SEIKO EPSON Corp., Nagano, Japan) and Leica M165FC (Leica Microsystems, Wetzlar, Germany). The proportion of the staining area to the total surface area was measured using Image-Pro Plus software (Media Cybernetics Inc., Rockville, MD, USA).

### 2.9. Statistical Analysis

All experiments in the present study were performed three times to test the reproducibility of the results, and representative findings are shown. In each experiment, at least three specimens were investigated for each condition. Levene’s Test for Equality of Variances using SPSS software was used to evaluate the normality of the data before choose the one-way analysis of variance (ANOVA). All values were expressed as the mean ± standard error of the mean. All results were compared using a one-way ANOVA followed by Tukey’s test with SPSS Statistics 17 (IBM, Armonk, NY, USA). Differences with a *p* values < 0.05 were considered to be significant.

## 3. Results

### 3.1. Microperforated Ti Membrane after MPNP Processing

A uniform topography of a micrometer scale was created on the material by the machining work. Figure 3 shows SEM images of a microperforated Ti membrane after MPNP processing. The perforated aperture ratio was very high and almost complete penetration occurred. The shape of each hole was precisely uniform at 25 × 25 μm. All of the holes were arrayed at uniform intervals of 75 μm. The surface of the head side which was impacted by the MPNP tool and pierced by needles, remained smooth and an array of square holes was precisely formed. The cut edges of apertures was straight and sharp (Figure 3a,c). On the other hand, the shape of each hole on the aperture at the tail side was generally uniform at 25 × 25 μm, the penetrated part of the substrate was accompanied by irregularly shaped edges with burrs mimicking a “volcanic” shape (Figure 3b,d). The microperforation process with MPNP was successfully performed for the whole range of punching (Figure 3e).

### 3.2. Cultivation of the Osteogenic Cell Line on the Microperforated Ti Membrane

A previous study reported that the topography of a Ti membrane affects cell proliferation and differentiation [17]. Therefore, we herein examined proliferation on the surface of a Ti membrane. The morphology of MC3T3-E1 cells on the microperforated Ti membrane was assessed using SEM images. After 3 days of cultivation, the number of cells increased on the substrate, automatically clustered and were organized. Figure 4a,b show that cells seeded on the head side were more likely to migrate to and accumulate on the square pores. Furthermore, Figure 4c,d show that cells passed though the porous path and were distributed on the opposite side. On the other hand, Figure 4e,f show that cells seeded on the tail side were distributed on the surface, but were more likely to remain within pores on the opposite side (Figure 4g,h). Cells on the tail side adhered to burr edges around microperforated square holes on the surface with pseudopodia.

We compared the cells cultured on the blank scaffold and head side of the Ti membrane for 30 days of cultivation. A large number of cells settled down and demonstrated a stellate morphology on the surface of the seeding side on the blank scaffold (Figure 5a), whereas only a few cells remained on the opposite side (Figure 5b). Cells covered whole pores on the head side of the microperforated Ti membrane, on which just several crevices existed (Figure 5c). Furthermore, a number of cells were observed on the opposite side (Figure 5d). These results indicated that cells had access to a sufficient amount of growth medium through the 3-dimensional microperforated scaffold structure in order to retain cell viability.

We also investigated cell density on the surface of the Ti membrane 21 days after the induction of osteogenesis. Figure 6a shows that cell density was markedly higher on the blank scaffold than on the microperforated membrane (*p* < 0.01). Furthermore, cell density was higher on the tail side than on the head side (*p* < 0.01) (Figure 6b). A monolayer of cells adhered to the head side of the Ti membrane to form a uniform cluster on micropores.

### 3.3. Expression of Genes Involved in Osteogenic Differentiation

MC3T3-E1 cells express OPN and OCN as osteogenic differentiation markers [18]. We examined the effects of the surface topography of the microperforated Ti membrane on the gene expression of these markers. Figure 7a shows that the gene expression levels of OCN were higher in cells cultured with osteogenic medium for 14 days than in those without osteogenic medium. Furthermore, cells cultured for 14 days on the tail side of the microperforated Ti membrane had lower expression levels of OCN than those cultured on the blank scaffold and head side (*p* < 0.05). On the other hand, OCN expression levels in cells cultured for 28 days were higher without than with osteogenic medium. Increasing rates of expression were higher on the microperforated Ti membrane than on the blank scaffold (*p* < 0.05). OCN gene expression levels in cells on the Ti membrane markedly increased, particularly after 28 days. OCN expression levels were higher on the head side than one on the blank scaffold and tail side in every conditions (*p* < 0.05). OPN expression levels in cells cultured on the surface of the microperforated Ti membrane were higher than those on the blank scaffold (*p* < 0.05) (Figure 7b). A stimulation with osteogenic medium decreased OPN expression levels in cells cultured for 28 days.

### 3.4. Evaluation of Calcification

We estimated the capability to induce the deposition of minerals on the topography of the microperforated Ti membranes. Figure 8a,b show that calcified deposits formed on all surfaces. On the head side, the rate of calcification was lower inside than outside the microperforated area. Mineral formation was lower on the microperforated area of the head side than on the tail side and blank scaffold (*p* < 0.001). The distribution of calcified deposits on the tail side was similar to that on the blank scaffold. Calcified areas were quantified in Figure 8c, which showed the proportion of the calcified area in the microporous section of specimen.

### 3.5. Protein Expression Involved in Osteogenic Differentiation

We investigated the expression of OPN and OCN at the protein level in cells cultured on the microperforated Ti membrane. The expression levels of OPN and OCN in MC3T3-E1 cells cultured for 7 days were negligible (Figure 9a). Figure 9b shows the expression of osteogenic differentiation markers in cells cultured for 14 days. OCN and OPN were initially expressed on day 14. The expression levels of OPN and OCN were higher in cells cultured with than without osteogenic medium (data not shown). Moreover, their expression levels were higher on the head side than on the tail side of the microperforated Ti membrane. Figure 9c shows that the expression levels of OCN and OPN were higher in cells cultured for 21 days than in those cultured for 14 days. OPN expression levels were lower on the tail side than on the head side, whereas OCN expression levels were higher on the tail side than on the head side. Between days 7 and 14, cells surviving on the head side uniformly populated the pores on the surface. As the number of cells increased over time, the population of MC3T3-E1 cells became homogenous on the Ti porous surface. However, cells on the tail side showed similar population characteristics to those on the blank scaffold.

## 4. Discussion

Previous studies investigated the influence of a micron-order surface topography of a culture substrate on the adhesion, migration, and proliferation of cells. Cells sense the status of substrate material after attachment and act in a manner of increasing their survival [19]. Factors controlling cell migration and proliferation on the material include the three-dimensional environment provided by the surface topography structured in a nano-meter to micro-meter scale [20,21,22]. Osteoblasts may be affected by the surface topography of biomaterials, such as pure Ti and hydroxyapatite, for the progression of proliferation and differentiation [23,24,25].

Surface modifications using laser processing techniques are widely employed for the addition of small pores or grooves to biocompatible substrates [26]. In relation to the medical applications of bone regeneration therapy, a thin microperforated pure Ti membrane was fabricated using a laser piercing method with a precise array of 20 μm holes that promoted the migration and proliferation of cells [27]. These morphological conditions of a micrometer scale on the surface of Ti have been utilized in osteoconductive therapy to promote bone regeneration [28]. However, in the laser processing of Ti, degeneration due to heat may reduce the quality of the surface, resulting in the deposition of metal dross around the cut edge subjected to laser ablation [27,29]. Although, a bulk of Ti that is relatively difficult to be applied a machining process for fabrication [30], it can be processed to precisely form a micropiercing array with cold working when the material is a sheet with a thickness of less than 10 μm [31]. The cold working for Ti may prevent thermal degradation and the deposition of dross, so the process of fabrication from Ti sheet using MPNP tool does not change the chemical properties of the material. On the other hand, the geometry of the material may also affect its hydrophilicity, which in turn, affects cell adhesion [12,32]. The present results indicated that the surface hydrophilicity of a material affects cell adhesion in cultivation. Therefore, to improve the hydrophilicity of the material examined, plasma technological processing was performed on the surface of the Ti membrane. Square holes with sides of 25 μm were formed at intervals of 75 μm, which reduced the surface area of the test material by 1/9 that of the control material. Therefore, flat areas were mostly preserved in the test material. Any defects on the flat surface of a substrate may have a negative impact on the adherence of and interactions between cells being cultivated. Cells migrate into micropores when cultivated, and are more likely to localize around square holes. Therefore, increases in cell numbers may be slightly smaller on the flat surface areas on the head side than on a control material [33]. In our previous studies on the functions of preosteoblast MC3T3-E1 cells cultured on a microporous Ti membrane, cells maintained basic survival in growth medium supplemented with 2% FBS, and the growth rate was very slow, which was suitable for long-term observations of the morphological structure of cells (30 continuous days) by scanning electron microscopy (SEM). Growth conditions with 5% FBS were shown to be beneficial for evaluating the osteogenic differentiation ability of MC3T3-E1 cells [34,35].

Even in the same material of Ti, the differences in topographies of the head and tail sides (Figure 3) on a micron scale altered the retaining capacity of the cell population in the cultivation. The area of the tail side was increased by the burr protrusions, and the nano-ordered roughness, that formed at the edge of the burr, was advantageous for cell attachment by entangling the pseudopodia of cells. A three-dimensional microstructure is included to increase the cell anchoring sites on the culture substrate and enhance cell attachment, thereby promoting cell body extension and migration. The maximum cell number capacity on the surface of the head side was lower than that on the tail side. However, the attachment of cell groups to the inside of the holes and in the vicinity of the apertures of holes suggested a stable environment for housing. The results of gene expression (Figure 7) indicated that the conditions under which cells accumulated on the microwell-like topography of the culture substrate influenced their differentiation.

Previous studies reported the expression of OPN at the early stage of osteogenic differentiation and that of OCN at a later stage than OPN [17,18]. The osteogenic differentiation markers were preferentially expressed with the accumulation of cultured cells. This is because the cell density was positively correlated with the expression of the markers on the test material. The structure of the head side without burrs facilitates the migration of MC3T3-E1 to the pathway of porous hole, even go through to the tail side (Figure 4). Accordingly, substrate morphology including the pathway of pores appeared an influence on the gene expression and production of the correspondent protein of the cells on the surface of both sides. This possible effect of communication of the cells between the head and tail might sometimes exchange a balance of the expressions in Figure 9.

A controlled rough surface on pure Ti promotes cell attachment and proliferation [36,37]. A micron-ordered regular topography formed with nano-level precision on biocompatible materials, including our prototypes, may promote a spatial directionality to the migration of cells accommodated on the substrate [38]. In the present study, the potential to form a precise and uniformed structure in a micron order to induce the differentiation of osteogenic cells was investigated. However, the effects of the geometrical properties of the microperforated Ti membrane on individual cultivated cells need to be quantitatively examined prior to its medical application to osseoinduction therapy.

Generally, from the viewpoint of standard specifications of the fine machining, burrs remaining on the workpiece after processing need to be removed. However, the surface topography of the tail side of the membrane did not appear to have a negative impact on cells. At least in in vitro cultivation, the micro-burrs made by machining processing of the surface of may support the migration and differentiation of the cells. Calcified nodules formed adjacent to the burrs around the apertures of holes. The surface topography in the micron order, including the roughness of burrs on the tail side, may be useful for inducing osteogenic differentiation.

We previously reported the biomedical application of a microporous Ti membrane fabricated using high-precision laser processing as a retainer of autologous and artificial bone on a recipient site in guided bone regeneration therapy [39]. The Ti membrane developed in the study was biocompatible and might induce the expression of osteogenic markers. A high-precision microtopography, created on the biocompatible cell culture substrate, may enable the control of osteogenic differentiation by adjusting cell distribution depending on the microtopographic structure.

The present study of a novel Ti membrane fabricated in the machining process with MPNP was able to provide precise arrayed holes in micron order. We focused on the performance of the unified microtopography on the membrane to influence to the growth and differentiation of osteogenic cell line in vitro. There were study limitations of the material due to select a single design that the dimension and shape could be precisely controlled on the micron order. Further research is needed to improve the surface topography promote osteogenic differentiation. Moreover the investigation of the performance of other tissue cells adjacent to the bone tissue such as fibroblasts, periosteal and periodontal ligament cells on the Ti membrane is required. The function of scaffold retaining or promoting the viability of osteogenic cells is crucial in the jaw bone regeneration therapy. The contribution of the Ti membrane as the scaffold for bone regeneration therapy should be developed by improvement of machining process optimizing the microstructure to the recipient site [40].

## 5. Conclusions

The present results demonstrated that a novel microporous Ti membrane fabricated using a precise mechanical punching process has the potential to facilitate the induction of preosteoblast cells into early osteogenic differentiation after a 14 days cultivation. The surface structure of the Ti membrane appears to affect the migration of MC3T3-E1 cells. In summary, the present study showed that the microporous Ti membrane that retained Ti primordial properties may be used to markedly stimulate the functionality of MC3T3-E1 in the differentiation stage.

## Figures and Tables

**Figure 1 materials-13-05288-f001:**
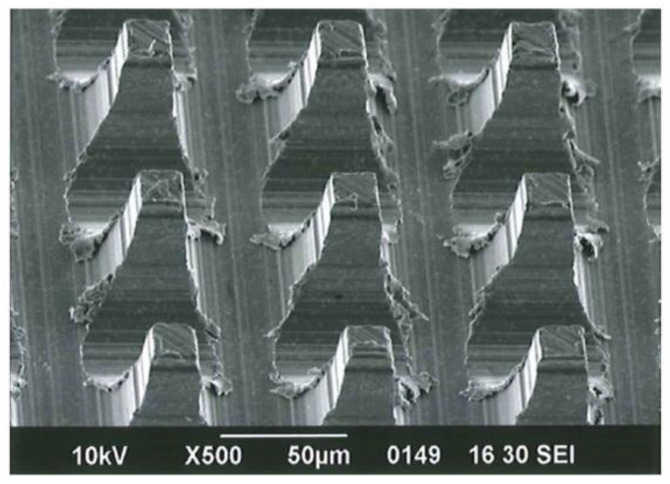
Scanning electron microscopy image of the microarrayed pin needle punch structure (JSM-6390LV, JEOL, Tokyo, Japan; scale bar represents 50 μm).

**Figure 2 materials-13-05288-f002:**
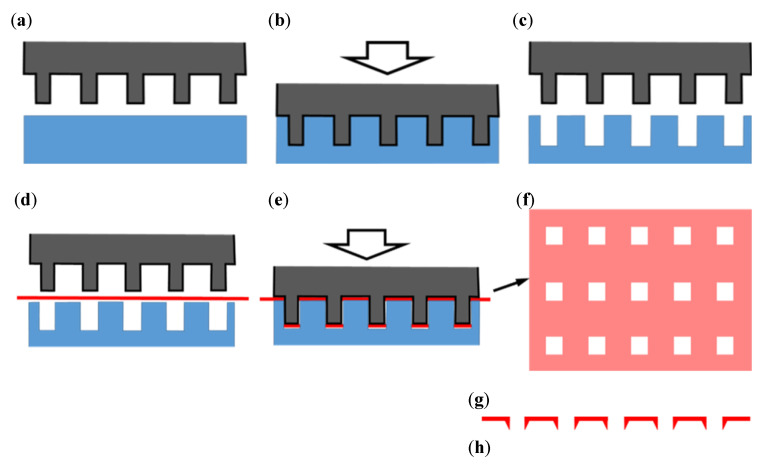
Processing steps for the microperforation of Ti sheets. (**a**) Fix a plastic resin plate on the workbench; (**b**) Compress MPNP into the plate; (**c**) Deform a die customized for engaging with MPNP; (**d**) Place a titanium sheet between MPNP and the die; (**e**) Punching; (**f**) The fabricated item after processing; (**g**) Head side; (**h**) Tail side.

**Figure 3 materials-13-05288-f003:**
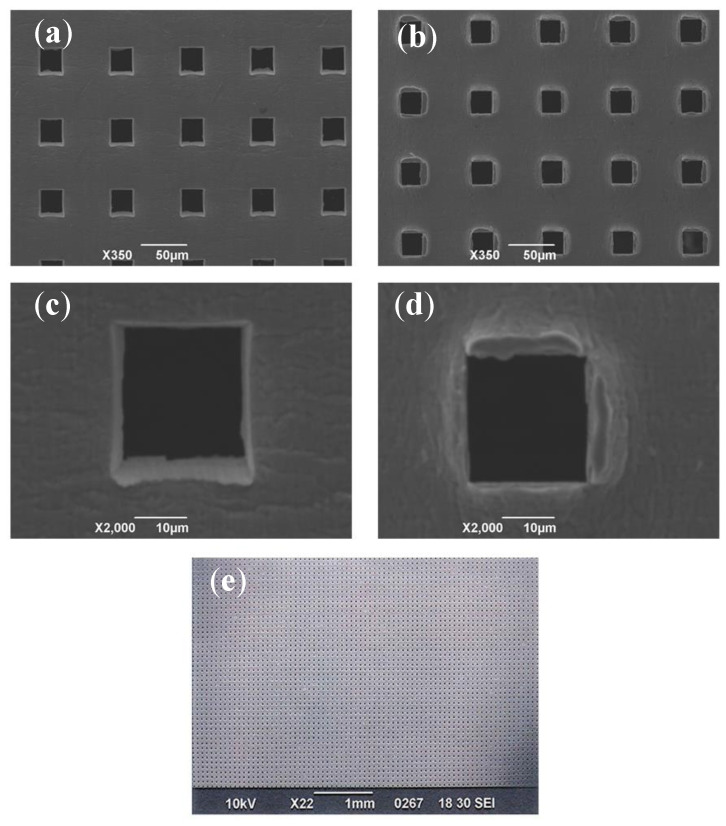
Surface structures of cell–free microperforated Ti templates using SEM. (**a**) An overview of 25 × 25 μm square pores, called the head side (scale bar represents 50 μm). (**b**) An overview of the tail side (scale bar represents 50 μm). (**c**) A higher magnification view of a pore on the head side (scale bar represents 10 μm). (**d**) A higher magnification view of the tail side (scale bar represents 10 μm). (**e**) A wide view of the template (scale bar represents 1 mm).

**Figure 4 materials-13-05288-f004:**
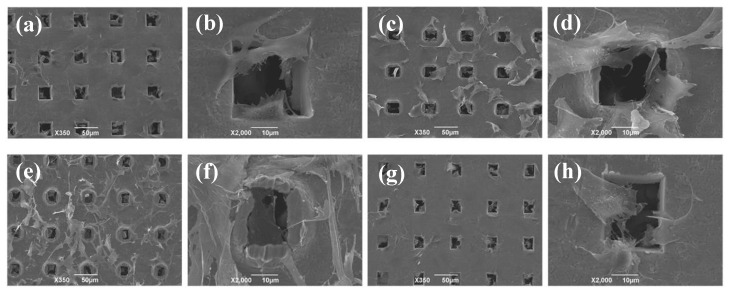
SEM images of MC3T3-E1 cells seeded on the Ti membrane on day 3. (**a**) MC3T3-E1 cells were seeded on the head side. (**c**) The opposite side of (**a**). (**e**) MC3T3-E1 cells were seeded on the tail side. (**g**) The opposite side of (**e**) (scale bar represents 50 μm). (**b**,**d**,**f**,**h**) are higher magnification views of (**a**,**c**,**e**,**g**) (scale bar represents 10 μm).

**Figure 5 materials-13-05288-f005:**
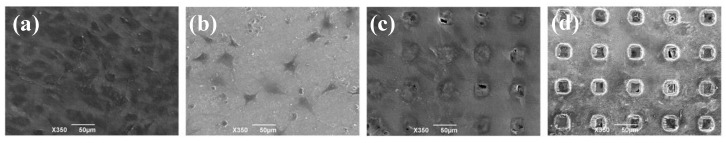
SEM images of MC3T3-E1 cells seeded on the blank scaffold and titanium membrane for 30 consecutive days. (**a**) MC3T3-E1 cells were seeded on the blank scaffold. (**b**) The opposite side of (**a**). (**c**) MC3T3-E1 cells were seeded on the head side of the Ti membrane. (**d**) The opposite side of (**c**) (scale bar represents 50 μm).

**Figure 6 materials-13-05288-f006:**
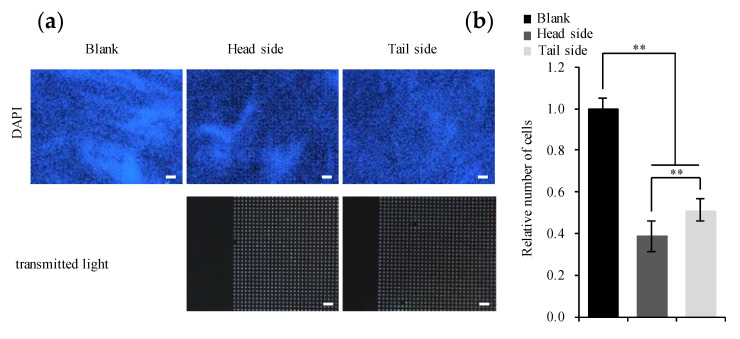
Topography of the Ti surface and effects on the proliferation of MC3T3-E1. Cells were incubated in α-MEM (5% FBS) with osteogenic supplement for 21 days. (**a**): Nuclei were visualized by staining with DAPI (blue). Scale bars represent 250 μm. (**b**): Stained areas in panel (**a**) were quantified, and relative expression was plotted by setting the value of DAPI in blank as 1. Representative data of three separate experiments are shown as the mean ± standard deviation of sextuplicate assays. The significance of differences is shown (** *p* < 0.01).

**Figure 7 materials-13-05288-f007:**
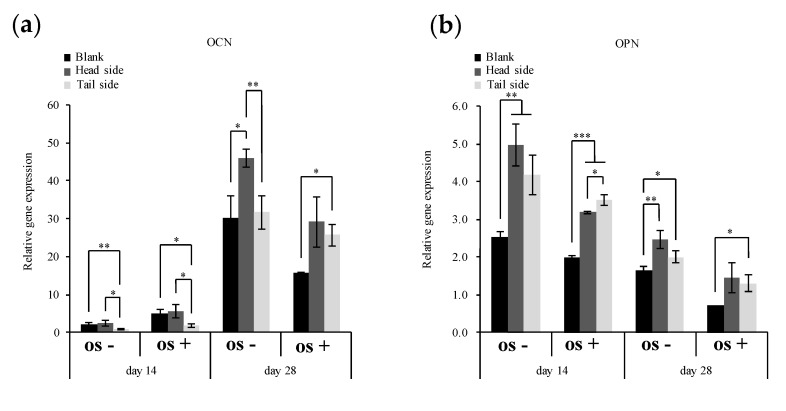
Relative gene expression of OCN (a) and OPN (b) on day 14 and 28. MC3T3-E1 cells on specimens were cultured in basal and osteogenic medium (OS). Real-time PCR data were calculated from independent samples (*n* ≥ 3). All values are shown as the mean ± standard deviation (*n* ≥ 3). All results were compared by a one-way ANOVA with Tukey’s test using SPSS Statistics 17. The results with *p* values < 0.05 were deemed to be significant (* *p* < 0.05, ** *p* < 0.01, *** *p* < 0.001); n.s., not significant (*p* > 0.05).

**Figure 8 materials-13-05288-f008:**
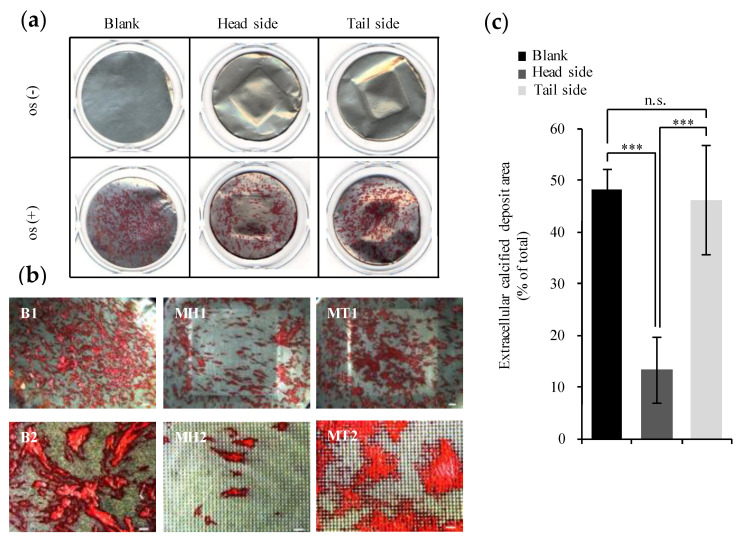
Calcification of osteoblasts on specimens after a 28 day induction of osteogenic differentiation. (**a**) Calcified deposits stained with ARS solution. (**b**) Surface appearance of calcified deposits on specimens using a biological microscope. A lower magnification view of calcified deposits (B1, MH1, and MT1). Scale bar represents 1000 μm. A high magnification view of calcification (B2, MH2, and MT2). Scale bar represents 250 μm. (**c**) Proportion of the calcified area in the microporous section of specimen (blank group as control). All points were calculated by Image-Pro Plus. Calcified deposits stained crimson. All values are shown as the mean ± standard deviation (*n* ≥ 3). All results were compared by a one-way ANOVA with Tukey’s test using SPSS Statistics 17. Results with *p* values < 0.05 were deemed to be significant (*** *p* < 0.001); n.s., non-significant (*p* > 0.05).

**Figure 9 materials-13-05288-f009:**
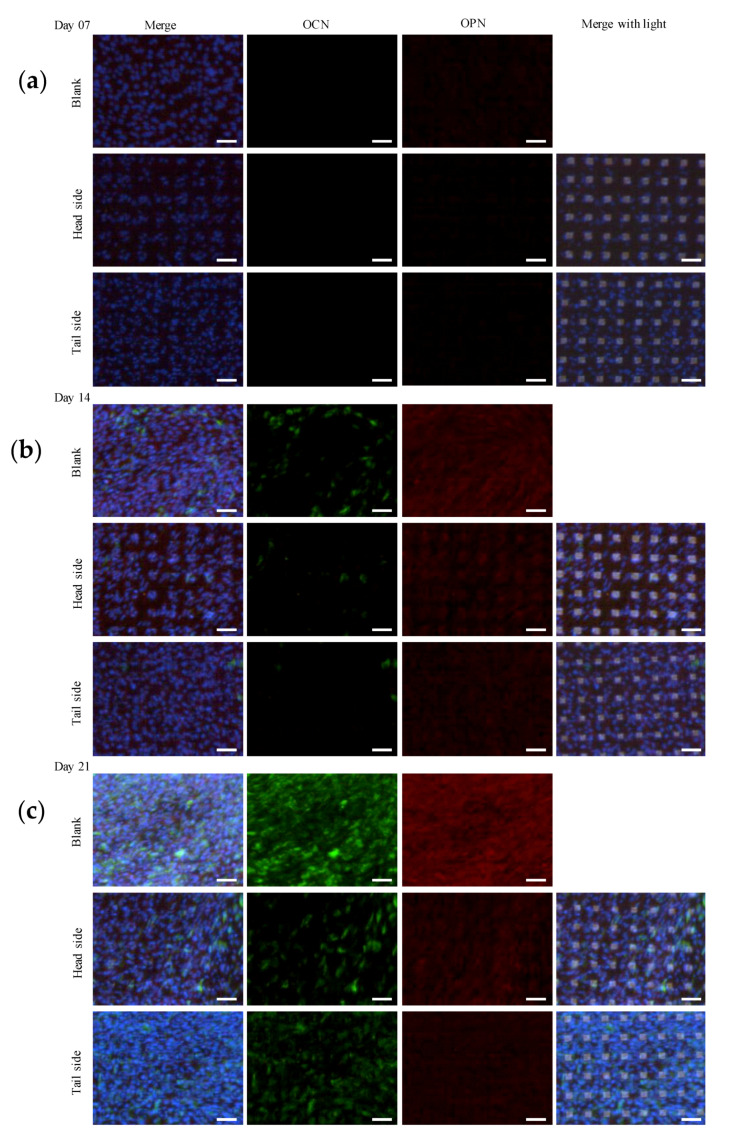
Immunofluorescence images of Ti surface-related OCN and OPN expression in MC3T3-E1 cells. Cells were incubated in α-MEM (5% FBS) with an osteogenic supplement for 7, 14 and 21 days. Nuclei, OCN and OPN were visualized by staining with DAPI (blue), Alexa Fluor^®^ 488 (green) and Alexa Fluor^®^ 555 (red). Scale bars represent 100 μm. (**a**–**c**) were for 7, 14, and 21 days respectively.

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
