# Peer review of "Evaluation of Preosteoblast MC3T3-E1 Cells Cultured on a Microporous Titanium Membrane Fabricated Using a Precise Mechanical Punching Process"

_materials, 2020, doi:10.3390/ma13225288_

Round 1

Reviewer 1 Report

1. Abstract, lines 31-31: this study did not evaluate bone regeneration, instead cell proliferation and activity. Please rewrite this sentence.

2. Lines 41-44: I do not understand what the authors mean with this sentence. Ceramics and polymers can also be manufactured in different shapes accordingly to their intended use. Please clarify.

3. Lines 58-60: what do authors mean by: “The  microtopographical surface of Ti produced by this process may sufficiently mimic the microstructure of MC3T3-E1 cells [12].” Do authors mean the tissue matrix where the cells develop? Please clarify.

4. Lines 57-65: this information presents a mix of material and methods and results. I suggest this to be removed and to be replaced by the study aim and/or hypothesis. What was the study's aim? To understand if this manufacturer process could produce Ti membranes with enhanced osteoblasts stimulation properties?

5. Lines 112: please clarify how the cells were seeded on both sides.

Which was the cell density at seeding?

6. Lines 117-124: please indicate at which time-point the cells were observed for cell morphology with SEM

7. Line 169: the authors refer here to results from at least three independent specimens. What does this mean? Three independent experiments, each one with one specimen?

8. Statical analysis: how was the normality of the data evaluated in order to choose the appropriate statistical test?

9. Figure 3: letters a, c, and e are missing from the figure. Also, please present the letters in the same position in all figures

10. Figure 3: letter f is missing

11. Figure 4: please correct the information on the x-axis

12. Figure 8 and 3.4 evaluation of calcification: the authors determined the extracellular calcified deposit in the 22 mm circle. However, on the head side, it is clearly visible an increased calcification around the microporous area and not in this area, which may have influenced the obtained results. What do the authors think about this?

13. I suggest the discussion section to be more focused and structured. The authors discuss the same topic in different sentences and repeat information.

14. Please add a paragraph regarding the study limitations to the discussion section

15. Lines 386-388: please remove this information since this was not evaluated

Author Response

Thank very much for your constructive suggestion to our manuscript. We are pleased to revise it according to the comments. We will provide the response as PDF file. 

We appreciate your kindly help for us.

Hiroshi Ishihata

Reviewer 2 Report

Dear Authors,

I have read the manuscript with interest and some questions raised. Attached please find my comments.

Overall. General English grammar revision (Minor spelling errors).

Abstract. Please add statistical tests.

Key words. “titanium tophography” and “Dentistry” could be added in my opinion.

Introduction. Authors stated “Furthermore, Ti or Ti alloy scaffolds, which are transplanted onto recipient sites, have high weight tolerance, corrosion resistance, and fatigue deformation”. Please add a reference for this statement.

Introduction. Authors stated “Novel dental implant products employ surface modifications on a microtopographical scale, even a nanotopographical scale, which promote the attachment and migration of cells from the surrounding living tissue at the recipient site”. Bedfore this sentence it should be pointed out the uses of titanium in Dentistry. It could be added that “Titanium is currently used in dentistry for implants (Modifications of Dental Implant Surfaces at the Micro- and Nano-Level for Enhanced Osseointegration. Yeo IL. Materials (Basel). 2019 Dec 23;13(1):89.) and miniscrews (Failure load and stress analysis of orthodontic miniscrews with different transmucosal collar diameter. Sfondrini MF, Gandini P, Alcozer R, Vallittu PK, Scribante A. J Mech Behav Biomed Mater. 2018 Nov;87:132-137) showing excellent mechanical and biological properties”.

Materials and Methods. Authors stated “On the plane surface of a metal block of a cemented carbide blank (Super ultra-fine grained carbide "AF1": Sumitomo Electric Hardmetal)…”. Please add City and State of the Manufacturer.

Materials and Methods. Authors stated “grooves with a width of 50 μm were serially carved by a grinding process at a precise equidistant of μm using a diamond dicing wheel”. Please add Model manufacturer, City and State of the wheel.

Materials and Methods. Authors stated “This process was performed using a fully automatic dicing saw and produced lattice arrayed needle punches that were precisely formed as uniform prism-shaped pins and the square side at the tip of the pins had a length of 25 μm”.  Please point out how the measurement was conducted. Additionally, please add the SEM model, Manufacturer, City and State.

Figure 1. Please enlarge a bit the image in order to improve readability.

Materials and Methods. Authors stated “They were incubated with PBS containing 0.25% Triton X-100 (PBS-T) for 10 min for permeabilization and then incubated in PBS-T with 1% (w/v) bovine serum albumin (Sigma) for 30 min to block non-specific binding”. Please add a reference for this statement.

Materials and Methods. Authors stated “Specimens were rinsed repeatedly with distilled water to remove excess dye. Images of stained cells were obtained with Epson ES-2200 (SEIKO EPSON Corp., Nagano, Japan) and Leica M165FC (Leica Microsystems). The proportion of the staining area to the total surface area was measured using Image-Pro Plus software (Media Cybernetics)”. Please add City and State of the manufacturers.

Materials and Methods. Authors stated “All results were compared using a one-way analysis of variance (ANOVA) followed by Tukey’s test”. ANOVA is performed with Gaussian distributions. Please state how normality of data was tested.

Materials and Methods. Authors used SPSS Statistics 17. Please add Software house, City and State.

Figure 3. Please enlarge a bit the image in order to improve readability.

Results. Section 3.2, 3.3 and 3.4. Please add p values when describing the results.

Figure 4. Please enlarge a bit the image in order to improve readability.

Discussion. Please add a paragraph concerning the limitations of the present report.

References. Year is lacking in reference 10

References. Some references are quite old (1983; 1999; 1992; 1997; 1998). If possible please switch with some more modern research. Some recent studies have been suggested in the sections above.

Author Response

Thank very much for your constructive suggestion to our manuscript. We are pleased to revise it according to the comments. The PDF file including a response for the review comments is attached.

We appreciate your kindly help.

Sincerely,

Hiroshi Ishihata

Round 2

Reviewer 1 Report

The performed corrections increased the manuscript quality. However, I still have minor suggestions for the manuscript.

Abstract lines 27-29 and section 3.5: The authors state that: “Osteocalcin expression levels were higher on the head side than one on the blank scaffold and tail side (p < 0.05). Osteocalcin protein expression levels were higher on the tail side than on the head side after 21 days of cultivation, and were comparable to the proportion of the calcified area (p < 0.05).” This result should be discussed, and a possible explanation for this be added to the manuscript. Why the gene expression is higher on the head side, and the correspondent protein is higher at the tail side? Is it related to the substrate morphology? What are the implications of this result?

Lines 63-65: please improve the English language in this sentence

Lines 165-166: please improve the English language in this sentence. I think the authors mean “evaluate” instead of “make”

Author Response

Thank you very much for your fast reply and helpful suggestion. We have revised the manuscript below:

Q: Abstract lines 27-29 and section 3.5: The authors state that: “Osteocalcin expression levels were higher on the head side than one on the blank scaffold and tail side (p < 0.05). Osteocalcin protein expression levels were higher on the tail side than on the head side after 21 days of cultivation, and were comparable to the proportion of the calcified area (p < 0.05).” This result should be discussed, and a possible explanation for this be added to the manuscript. Why the gene expression is higher on the head side, and the correspondent protein is higher at the tail side? Is it related to the substrate morphology? What are the implications of this result?

A:Thank you very much for your constructive suggestion. We all agree with you that the morphological feature of the test material may produce antinomy of results on the gene expression and correspondent protein. We found that the osteocalcin expression levels were higher on the head side than one on the blank scaffold and tail side (Fig. 7). However, the level of calcified nodule generally showed a positive correlation with the cell density on the substrates (Fig. 8). The osteogenic differentiation markers were preferentially expressed with the accumulation of cultured cells on the head side of test material. This is because the cell density was positively correlated with the gene expression of the markers. The structure of the head side without burrs facilitates the migration of MC3T3-E1 to the pathway of porous hole, even go through to the tail side (Fig. 4). Accordingly, substrate morphology including the pathway of pores appeared an influence on the gene expression and production of the correspondent protein of the cells on the surface of both sides. This possible effect of communication of the cells between the head and tail might sometimes change a balance of the expressions in the immunofluorescence images (Fig. 9).

The some part of the explanation above was appeared to the discussion part of lines 337-342.

Q: Lines 63-65: please improve the English language in this sentence.

A: We erased the one of the sentence and corrected.

Lines 165-166: please improve the English language in this sentence. I think the authors mean “evaluate” instead of “make”

A: Corrected.

Reviewer 2 Report

good job

Author Response

Dear Sir:

Thank you very much for your kindly help.

Sincerely,

Hiroshi Ishihata